# Tailoring the Diagnostic Pathway for Medical and Surgical Treatment of Uterine Fibroids: A Narrative Review

**DOI:** 10.3390/diagnostics14182046

**Published:** 2024-09-14

**Authors:** Gabriele Centini, Alberto Cannoni, Alessandro Ginetti, Irene Colombi, Matteo Giorgi, Giorgia Schettini, Francesco Giuseppe Martire, Lucia Lazzeri, Errico Zupi

**Affiliations:** Department of Molecular and Developmental Medicine, University of Siena, 53100 Siena, Italy; albertoacannoni@gmail.com (A.C.); ginettialessandro14@gmail.com (A.G.); colombi.irene1@gmail.com (I.C.); matteogiorgi27@gmail.com (M.G.); giorgiaschettini@gmail.com (G.S.); francescogmartire@libero.it (F.G.M.); lucialazzeri79@gmail.com (L.L.); errico.zupi@unisi.it (E.Z.)

**Keywords:** leiomyoma, diagnosis, fibroid, medical therapy, gynecological surgery, ultrasound, hysteroscopy, robotic surgery

## Abstract

Uterine leiomyomas are the most common benign uterine tumors in women and are often asymptomatic, with clinical manifestation occurring in 20–25% of cases. The diagnostic pathway begins with clinical suspicion and includes an ultrasound examination, diagnostic hysteroscopy, and, when deemed necessary, magnetic resonance imaging. The decision-making process should consider the impairment of quality of life due to symptoms, reproductive desire, suspicion of malignancy, and, of course, the woman’s preferences. Despite the absence of a definitive cure, the management of fibroid-related symptoms can benefit from various medical therapies, ranging from symptomatic treatments to the latest hormonal drugs aimed at reducing the clinical impact of fibroids on women’s well-being. When medical therapy is not a definitive solution for a patient, it can be used as a bridge to prepare the patient for surgery. Surgical approaches continue to play a crucial role in the treatment of fibroids, as the gynecologist has the opportunity to choose from various surgical options and tailor the intervention to the patient’s needs. This review aims to summarize the clinical pathway necessary for the diagnostic assessment of a patient with uterine fibromatosis, presenting all available treatment options to address the needs of different types of women.

## 1. Introduction

Uterine leiomyomas, or fibroids, are the most common benign tumors affecting women of reproductive age. While they are clinically apparent in 20–25% of women, histological diagnoses after hysterectomy suggest a prevalence of up to 70%, with rates as high as 80% among black women by the age of 50 [1,2,3]. Despite their prevalence, many fibroids remain undiagnosed due to their asymptomatic nature, potentially leading to an underestimation of their true impact [4]. The substantial prevalence of uterine fibroids significantly impacts global healthcare expenditures. Research estimates indicate that annual direct and indirect costs associated with uterine fibroids reach $34.4 billion in the United States [5].

The pathogenesis of fibroids is multifactorial, involving genetic mutations (e.g., MED12), hormonal imbalances, and environmental factors that contribute to abnormal cell proliferation and excessive extracellular matrix deposition [6,7,8,9,10]. Fibroids often overexpress hormonal receptors for estrogen and progesterone, driving their growth in hormonally favorable environments [11,12,13]. Risk factors for fibroid development include early menarche, obesity, and chronic stress, while protective factors include multiparity and the use of combined oral contraceptives [14,15,16,17,18,19,20].

Symptomatically, fibroids range from being asymptomatic to causing significant morbidity, including abnormal uterine bleeding (AUB), pelvic pain, and reproductive challenges such as infertility and recurrent miscarriage [21,22,23,24]. These symptoms depend largely on the fibroid’s location—submucosal fibroids, in particular, are more likely to cause AUB and reproductive issues [25,26,27].

Given the widespread occurrence of fibroids and their potential impact on quality of life, accurate diagnosis and tailored management are essential. The diagnostic pathway often involves clinical examination, imaging techniques like ultrasound and MRI, and sometimes histological analysis to confirm the diagnosis and plan treatment [9,26]. Management strategies for fibroids are diverse, ranging from medical therapies that target hormonal pathways to surgical interventions like myomectomy and hysterectomy, depending on the size, location, and symptoms of the fibroids [7,23].

This review aims to provide a comprehensive overview of the clinical pathway for the diagnostic assessment of patients with uterine fibromatosis. It encompasses a step-by-step approach starting from initial clinical suspicion to advanced imaging techniques and diagnostic procedures. Furthermore, this paper presents a detailed evaluation of all available treatment options, including both conservative and surgical approaches, to address the diverse clinical presentations and needs of different patient profiles. Emphasis is placed on individualizing treatment strategies to align with the patient’s reproductive goals, symptom burden, risk of malignancy, and personal preferences, thereby optimizing both clinical outcomes and quality of life.

## 2. Diagnosis

### 2.1. Ultrasonographic Diagnosis: Transvaginal Sonography and Sonohysterography

The diagnosis of uterine fibroids is primarily based on imaging techniques. Frequently, myomas are detected incidentally during transvaginal sonography (TVS) in asymptomatic or symptomatic patients. Ultrasound is the first-line imaging technique in the diagnosis of uterine leiomyomas because of its wide availability, non-invasiveness, and low cost [28]. The aim of TVS, as well as other imaging techniques, is to detect the presence of myometrial lesions, to evaluate the dimensions and localization of each lesion and to assess their nature, excluding differential diagnoses such as adenomyosis, smooth tumors of unknown malignant potential (STUMPs) and leiomyosarcomas, although the final diagnosis is only anatomopathological. Depending on patient preference and uterine dimensions, ultrasound pelvic assessment can be performed using transvaginal or transrectal probes, or alternatively through transabdominal scanning, which is preferred in virginal patients or in cases of subserosal or voluminous fibroids extending beyond the small pelvis that otherwise cannot be well evaluated. The ultrasonographic characteristics of leiomyomas are well defined in the literature, and the first accurate description of the ultrasonographic features of fibroids was published in 2015 in the MUSA (Morphological Uterus Sonographic Assessment) consensus [29]. They appear as rounded solid lesions with well-defined margins and internal fan-shaped shadows. In some cases, calcifications may be present, appearing as hyperechoic areas within the lesion, or the internal content may be inhomogeneous due to lipomatous or ialine degeneration. The use of color or power Doppler gives additional information about the vascularization of the lesions, and usually myomas present a peripheral vascularization and a rare and scarce internal blood flow. This feature is of particular interest in the differential diagnosis of benign and malignant lesions, as the latter tend to have a central distribution of vessels even though there are some exceptions (Figure 1).

In a recent study, Russo et al. [28] put in correlation the ultrasonographic characteristics of highly vascularized uterine fibroids with their histopathological diagnosis. The authors found that of 70 fibroids with extensive central vascularization, 7% were sarcomas, highlighting that the vascular pattern is a strong risk factor for malignancy, but a central flow was also present in 93% of lesions of a benign nature making the differential diagnosis challenging and requiring additional criteria to determine the nature of the findings. Elements that need to be assessed include patient age, number of lesions, borders, and ultrasound features. Myomas generally have regular borders and are more likely to be multiple, while a single lesion with inhomogeneous echogenicity is more likely to be malignant. Cystic areas are confounding features as they can be present in both cases, and are presumed to be due to necrosis. Russo et al. [28] found this characteristic in 31.3% of typical leiomyomas, in 55.2% of leiomyoma variants as myxoid, and in 40% of malignant lesions. On the other hand, the presence of calcifications and shadowing are considered reassuring features, as they are generally not present in malignant lesions. In a recent study, De Bruyn et al. [30] confirmed that sarcomas more often show an irregular shape, with non-uniform echogenicity, cystic areas and necrosis, and are highly vascularized, while calcifications are generally absent, as described above. Once the presence of myomas has been identified, the second step is to correctly describe their position in the myometrium and their relationship to the endometrial cavity and serosa according to the FIGO classification system [21]. In the preoperative view, this aspect is fundamental for planning an accurate and tailored surgery. Submucosal fibroids (i.e., FIGO 0-1-2) require a hysteroscopic approach. In these cases, the intracavitary portion needs to be measured and further information on the myometrial-free margin to the serosa is useful for precise surgical planning. For this reason, two possible techniques can be added to 2D TVS: saline contrast sonohysterography (SHG) and 3D TVS, which are becoming valid alternatives to diagnostic hysteroscopy (Figure 2).

In symptomatic women, sonohysterography can be considered an equivalent diagnostic approach to hysteroscopy, particularly for the localization of submucosal uterine leiomyomas [31], with the advantage over 2D TVS of enhancing the contrast between the myometrium, fibroids, and uterine cavity, thereby helping to differentiate lesions. Sonohysterography involves the distension of the cavity with saline used as a contrast medium, enhancing the relationship between the myoma and the uterine walls and allowing an accurate measurement of the intracavitary part. Sonohysterography should be preferred over hysteroscopy as second-line imaging because it is less invasive, better tolerated, and allows assessment of the size, depth, and vascularization of the entire myoma, whereas hysteroscopy can only visualize the intracavitary portion of the lesion [31]. Three-dimensional TVS, when available, can be a useful tool to exactly locate and measure the myoma by obtaining a coronal view of the uterus, but the relationship with the uterine cavity is not as accurate as per SHG [32]. Furthermore, a 3D scan can be integrated with SHG; however, a recent review reported a sensitivity of 94.5% and a specificity of 99.4% for SHG, with no statistically significant differences between 2D and 3D SHG in detecting intracavitary lesions [33]. One aspect to consider is that fibroids can be associated with pain and AUB, but they can also be implicated in infertility through several mechanisms: they can lead to the distortion of the uterine cavity, alteration of endometrial and myometrial vascularization, interference with uterine contractility, the triggering of hormonal, paracrine and molecular alterations, and impairment of endometrial receptivity [34]. Subserosal fibroids are less likely to be symptomatic; however, they can have various clinical manifestations, potentially causing pelvic pain, urinary discomfort and frequent bladder voiding, venous thrombosis, and low back pain. The symptoms might be related to the specific localization and the size of the myoma. Two specific circumstances deserve specific attention because they may constitute a possible cause of acute pain: the torsion of a pedunculated myoma and the myoma’s necrosis. In the presence of pedunculated leiomyomas, torsion may be a possible consequence of the patient’s movements, although this is a rare complication. The diagnosis is not always simple because the pedicle flow of the myoma, examined through color Doppler, may not be absent [35]. On the other hand, the degeneration and necrosis of a myoma is usually the consequence of rapid growth, and it occurs in large myomas because of blood supply insufficiency in the center of the lesion. This condition typically occurs in large myomas during pregnancy where they are likely to grow up to 20% in volume in the first 16 weeks. A peculiar localization of myomas is between the sheets of the broad ligament or the parametrium; lesions with this characteristic are defined as infraligamental myomas (FIGO 8) [21,36]. Occurring in only 6% of cases, these fibroids are quite rare, but the diagnosis is important because their surgical removal can be challenging and is associated with a higher risk of ureteral and uterine artery injury. As these fibroids are often retroperitoneal, ureteral dissection and identification of the uterine artery at the origin are often necessary to avoid complications [36].

### 2.2. Hysteroscopy

Another diagnostic option for submucosal fibroids is hysteroscopy, which is considered the gold standard for evaluating the uterine cavity. The main advantage of this technique is that it is a tool that allows not only direct visualization of the lesion but also treatment of the disease; it is known as “see and treat” and can be carried out in an outpatient setting [37].

Adequate assessment of the lesions is essential in order to consider all the possible risks associated with the procedure, as well as the difficulty of the surgery, and the possibility of a second surgery. In order to plan an appropriate intervention, it is essential to know the extension of the intracavitary portion of the fibroid, as well as the intramural part and the myometrium free distance to serosa, which, as described above, can only be assessed by ultrasound since, in hysteroscopy, the fibroid appears as a solid, generally smooth protrusion of the uterine wall, but only the intracavitary extension can be visualized with this technique [38]. Ideally, hysteroscopy should be performed during the proliferative phase of the menstrual cycle, from day 4 to day 11, when the endometrium is thinner and allows better visualization of the cavity. The preferred technique in the outpatient setting is vaginoscopy, which reduces pain significantly, allows visualization of the vaginal walls and cervix, and facilitates gentle distension of the orifice by water [37]. This procedure is generally safe, but sometimes complications can occur. The most common of these is a vasovagal reaction with a prodrome of dizziness, diaphoresis, nausea, bradycardia, and pallor, which recovers after a few minutes following removal of the stimuli and, rarely, it may be necessary to administer 5 mg of atropine and oxygen. The most serious complication is uterine perforation, which is recognized by loss of uterine distention and sometimes visualization of the mesenteric adipose tissue. If no energy has been used and adjacent organs such as the bladder and rectum are not involved, observation is sufficient; otherwise laparoscopy may be required [37]. In the absence of anatomical markers or insufficient distension, a false passage must be suspected. The inflow and outflow of the distension solution must be accurately recorded to avoid the risk of intravasation.

### 2.3. Magnetic Resonance Imaging

Magnetic resonance imaging (MRI) is a radiation-free, high-resolution technique that can visualize the anatomical layers of the uterus and detect small lesions within these layers. It is generally not a first-line imaging technique (Table 1) and is only used in complex or doubtful cases [39]. The sagittal plane is the best section for detecting and localizing fibroids [40]. MRI uses different sequences to help recognize different aspects of lesions. On T1 sequences, non-degenerated fibroids and calcifications have a characteristic high T1 signal and appear as low to intermediate signal intensity in comparison with normal myometrium; red degeneration can be visualized as a T1 hyperintense rim around the fibroid. On T2 sequences, they generally show low signal intensity and hypervascularization. Cystic degeneration or necrosis is usually associated with a high T2 signal, whereas a low T2 signal is more often associated with hyaline degeneration [39]. As mentioned above, benign leiomyomas may occasionally present with degenerative changes, resulting in atypical imaging findings also on MRI, particularly heterogeneous high signal intensity (SI) on T2WI and/or intralesional hemorrhage, mimicking uterine sarcomas. To distinguish these ‘atypical leiomyomas’ from uterine sarcomas, several authors have observed that sarcomas with a lower apparent diffusion coefficient (ADC) value are associated with a higher likelihood of malignancy. In addition, ill-defined margins, intralesional hemorrhage, and the absence of T2 dark areas are usually present in sarcomas [41].

## 3. Medical Therapy

With respect to medical therapy, the options available for the clinicians to treat symptomatic fibroids are various. Medical therapy includes drugs aimed at the control of the symptoms—with a direct action on the endometrium—such as COCs and progestins, and drugs that detain an effect both on the endometrium and on the dimension of fibroids, such as GnRH analogs, GnRH antagonists, and Ulipristal acetate, a selective progesterone receptor modulator (SPRM). Recently, new GnRH antagonists (GnRHants) such as Elagolix, Relugolix, and Linzagolix have emerged, demonstrating excellent control of fibroid-related heavy menstrual bleeding (HMB) and the preservation of bone mineral density when combined with add-back therapy. These agents represent an alternative option for long-term management [42]. When choosing a medical therapy, a gynecologist should evaluate the primary objective of the patients and, after merging data coming from the imaging techniques, tailor a long-term strategy for the patient. The personalization of medical treatment depends on several factors, including the patient’s age, comorbidities, desire for pregnancy, need to preserve fertility, symptoms severity, suspicion of malignancy, and proximity to menopause. Pharmacological therapy often acts as a bridge before surgery, aiming to improve surgical outcomes by reducing symptoms and potentially shrinking fibroids. However, medical therapy can also be a definitive treatment for symptomatic premenopausal patients who are not eligible for surgery [43]. Despite the numerous therapeutic options available, if there is inadequate symptom control with medical therapy or the suspicion of malignancy, the treatment choice shifts to surgery or alternative techniques [44].

### 3.1. SPRMs (Selective Progesterone Receptor Modulators)

The action of SPRMs is expressed by their ability to bind to the progesterone receptor either in an agonistic, antagonistic, or mixed agonistic/antagonistic way [45]. On one hand, it is well known that progesterone plays a pivotal role in the growth process of fibroids, as it was highlighted in several in vitro studies [46]. On the other hand, other studies demonstrated the antiproliferative effect of SPRMs, with a proapoptotic effect aimed selectively at fibroid cells and a reduction in fibrotic activity and in growth factor expression [47].

#### 3.1.1. Mifepristone

Mifepristone (RU-486) has an exclusively antagonistic action on the progesterone receptor, and, as it emerged in a meta-analysis published in 2013, which included 11 RCTs where patients with symptomatic fibroids were treated with mifepristone, its use is associated with a volume reduction both in fibroids and in the uterus, and with a clinical improvement regarding symptoms [47]. In fact, mifepristone demonstrated its efficacy in reducing abnormal uterine bleedings (AUBs) and fibroid-related dysmenorrhea. Some authors proposed a low-dose mifepristone scheme (2.5 mg/die) for 3 to six months, whereas others studied the efficacy and safety of daily oral mifepristone therapy at higher doses (5 to 10 mg/die) for up to 12 months. The effect of 10 and 25 mg (oral, daily) mifepristone therapy has been compared to that of 3.75 mg Leuprorelin (subcutaneous, monthly) in an RCT, showing similar efficacy in the management of symptoms for a 3-month period of therapy [48,49].

#### 3.1.2. Ulipristal Acetate 

Ulipristal acetate (UPA) has a significant antiproliferative effect on UFs. Following the FDA approval obtained in 2012 of the PEARL I and PEARL II RCTs, a new groundbreaking possibility for the treatment of fibroids emerged [47]. In these trials, UPA showed optimal efficacy both in the management of AUBs and in the volume reduction of fibroids before surgical treatment. In addition, when compared with GnRH agonists in women, it was demonstrated to be non-inferior and less likely to induce amenorrhea-related symptoms, while having a faster effect in controlling abnormal uterine bleedings considering that up to 50% of patients treated with daily 5 mg UPA experience amenorrhea after only 10 days of treatment [50,51].

Two subsequent RCTs, PEARL III and PEARL IV, evaluated the efficacy of UPA in the long-term medical management of fibroids with outstanding results in terms of bleeding control (>80%) and size reduction—close to 50% [52,53].

Other potential advantages of the use of UPA before surgical treatment include the possibility of treating anemia, which frequently affects patients with symptomatic fibroids, and the fact that its pharmacological effects persist for up to six months after the end of the treatment, allowing the surgeon to postpone and tailor the subsequent surgical treatment to the needs of the patients [54]. As opposed to that, the use of GnRH agonists is associated with the so-called “rebound effect”, which indicates the immediate rebound in growth effect after the cessation of the therapy. The therapy using SPRMs is associated with a spectrum of endometrial changes called PAECs (PRM-associated endometrial changes) which are present in up to 30% of women undergoing therapy and include cystic glandular dilation and an asymmetry of growth in the endometrial stroma and epithelium with increased apoptotic activity [51]. These modifications, gathered together, constitute a histologic pattern that differs both from physiological endometrium and from endometrial dysplasia. However, it has been demonstrated that these endometrial changes tend to disappear after 3 to 6 months from the cessation of the therapy with a lower incidence of endometrial hyperplasia when compared to the control group (<0.9%) [55,56]. The most frequent side effects caused by UPA are, with their relative incidence, hot flushes (5.7%) breast tenderness (3.0%) and cephalgia (10.0%) [52]. In all the PEARL studies, estradiol serum levels were evaluated, the results showing them to be higher than menopausal ones, with no difference in bone mineral density (BMD) observed. Regarding coagulation, no changes were detected in patients under therapy [52]. After its commercialization, sporadic cases of hepatic toxicity associated with the use of UPA emerged, leading to the product’s withdrawal from the market in 2020 following an evaluation by the Pharmacovigilance Risk Assessment Committee (PRAC). In 2021, after a thorough review, the European Medicines Agency (EMA) reauthorized the use of UPA, restricting it to symptomatic premenopausal patients for whom surgical or alternative treatments were ineffective or contraindicated, provided that liver function was carefully monitored [57].

### 3.2. GnRH Analogs

GnRH analogs determine a “temporary menopause” with amenorrhea characterized by a hypo-estrogenic environment. The mechanisms of action of these molecules consist of the suppression of the pituitary–ovarian axis through receptor downregulation, following an initial flare up effect, which decreases the production of gonadotropins and gonadal steroids. GnRH analogs must be administered via intramuscular injection every 1 or 3 months, possibly leading to a problem of therapeutic compliance. Amenorrhea occurs in more than 98% of women, with a 35–65% reduction in fibroid size within 3 months of starting therapy [5,58]. This pharmacological effect determines an improvement both in hemoglobin serum levels in patients with anemia secondary to symptomatic fibromatosis and in terms of fibroid volume reduction [59]. In fact, GnRH analogs demonstrated their efficacy in the presurgical treatment of fibroids [60] and several studies have shown how the presurgical use of GnRH analogs, in addition to the aforementioned effects, decreases endometrial thickness and impairs fibroid vascularization, thus reducing surgical blood loss [61]. In a prospective randomized trial, Seracchioli et al. demonstrated that in patients with large uteri, preoperative treatment with GnRH agonists for 3 months could facilitate the hysterectomy procedure, reduce uterine size, decrease operative time, and reduce overall blood loss [62]. Because of the side effects associated with GnRH analogs, all linked with hormonal deficiency, the optimal duration of the therapy is 3 to 6 months, while the persistence of the effect, in terms of volume loss, is low, as it is demonstrated how fibroids can grow back, following the so-called “rebound effect”, in only 12 weeks after the cessation of the therapy [63]. The majority of symptoms experienced by patients under therapy with GnRH are related to the low estrogen levels, and include hot flashes, vaginal atrophy and/or dryness, sexual dysfunction, decreased libido, and osteoporosis [64]. To overcome this issue, which can lead to the abandonment of the therapy, it is possible to prescribe, concomitantly, an add-back therapy. In a prospective RCT focusing on long-term (>6 months) therapy with GnRH analogs, in association with combined oral contraceptives (COCs), it emerged that by combining GnRH analogs with COCs or progestins, the bone mineral density reduction remains stable, at 3% [65,66]. A systematic review showed how the use of tibolone and raloxifene as preventive tools against bone mineral density loss can be helpful during GnRH analog therapy [67].

### 3.3. GnRH Antagonists

Oral GnRH antagonists (GnRHant) competitively bind to GnRH receptors, leading to an immediate suppression of gonadotropin release without the “flare-up” effect seen with agonists, thereby rapidly inducing a reversible menopausal state. Available in oral formulations, Elagolix, Relugolix, and Linzagolix have demonstrated excellent control of heavy menstrual bleeding (HMB) associated with uterine fibroids. When combined with add-back therapy, they preserve bone mineral density, allowing these formulations to be used for long-term management [68]. Among the molecules belonging to this class is Elagolix, combined with estradiol and norethindrone, approved in 2020 for the treatment of HMB associated with uterine fibroids for up to 24 months. This oral therapy involves taking two tablets: one containing the add-back therapy to be taken in the morning, and the other containing Elagolix to be taken in the evening. Simon et al., in the phase 3 UF-EXTEND study, demonstrated that this treatment significantly reduces menstrual blood loss, with over 85% of participants experiencing substantial reductions, more than 50% achieving amenorrhea at 12 months, and an increase in hemoglobin levels by over 2 g/dL at 6 months compared to placebo. The introduction of add-back therapy effectively limited the side effects related to hypoestrogenism, with minimal impact on bone mineralization [69]. In May 2021, a once-daily treatment with Relugolix (40 mg) combined with add-back therapy (estradiol 1 mg and norethindrone acetate 0.5 mg) was approved for the treatment of HMB associated with uterine fibroids for a period of 24 months. According to data from the LIBERTY studies, the combination therapy with Relugolix demonstrated a rapid and consistent reduction in menstrual blood loss (over 70% of patients achieved the primary endpoint at 24 weeks), an increase in hemoglobin levels, a higher rate of amenorrhea, and a slight decrease in fibroid volume. The treatment is generally well tolerated with few side effects, and notably, bone mineral density (BMD) was preserved over the 2-year treatment period [70]. In June 2022, Linzagolix was approved as a treatment for fertile-age patients with moderate-to-severe symptomatic uterine fibroids. As an oral GnRH antagonist, Linzagolix is available in once-daily doses of 100 mg or 200 mg. Studied both with and without add-back therapy, Linzagolix has shown significant reduction in uterine fibroid-related menstrual bleeding. The lower dose (100 mg) partially suppresses the hypothalamic-pituitary–ovarian axis, while the higher dose (200 mg) induces complete suppression. Both doses of Linzagolix, with or without add-back therapy, have demonstrated a substantial reduction in bleeding. Add-back therapy improves reductions in bone mineral density. A daily administration of 100 mg of Linzagolix as a monotherapy may be considered for long-term treatment in cases where additional hormonal therapy is not feasible. Alternatively, higher doses as a monotherapy may be considered when the primary goal is cessation of bleeding and the correction of anemia [71].

### 3.4. Combined Oral Contraceptives (COCs)

Combined oral contraceptives (COCs) are commonly employed in clinical practice to manage fibroid-related bleeding. Their efficacy to control abnormal uterine bleeding is limited, and no modifications on fibroid volume were ever documented in the literature [72]. In the past, COCs were thought to be a risk factor for the growth of uterine fibroids due to the stimulatory effects of both estrogens and progestins. However, a recent meta-analysis suggests that the presence of uterine fibroids should not be considered a contraindication for the use of COCs. Evidence from epidemiological and clinical reviews suggests that both combined oral contraceptives and progestin-only contraceptives could lower the risk of developing clinically significant uterine fibroids [73]. Regarding bleeding control, an RCT compared the efficacy of COCs and oral medroxyprogesterone on reducing acute uterine bleeding in non-pregnant gestational-age women, showing how both treatments were effective, with COCs daily administration reducing bleeding up to 88% in a median time of three days [74]. Another small observational study compared COCs and placebo, highlighting a statistically significant reduction in bleeding in the COCs group without any variation in fibroids volume [75]. Compared with levonorgestrel-releasing intrauterine system (LNG-IUS), COCs were less effective in reducing menstrual bleeding in patients with fibromatosis [76]. However, COCs represent a very versatile and manageable type of drug and are largely employed by clinicians both as a tool to temporarily control abnormal uterine bleeding and as a bridge therapy to menopause in perimenopausal patients.

### 3.5. Oral Progestins and Levonorgestrel-Releasing Uterine System (LNG-IUS)

Progestins have a well-known, wide spectrum of pharmacological effects on the uterus, causing endometrial atrophy while having a dual effect on fibroid growth. Progestins stimulate fibroid cell proliferation through the increase of the expression of Epidermal Growth Factor (EGF), whereas they inhibit proliferation by down-regulating insulin-like growth factor-1 (IGF-1), estrogen and progesterone receptor [77]. A study on the menstrual patterns of women with or without uterine fibroids treated with LNG-IUS showed how this therapy can reduce significantly both blood loss and uterine volume while not producing any variation on fibroid volume [78]. LNG-IUS, by acting locally on the endometrium with minimal systemic absorption and minimal adverse effects, can induce amenorrhea and/or improve menorrhagia and anemia in up to 50–60% of patients with abnormal uterine bleeding (AUB) within 6–12 months, including women with fibroids. However, it should be noted that several studies have reported higher rates of device expulsion in women with fibroids, especially with fibroids larger than 3 cm, compared to women without fibroids (6–12% vs. 0–3%) [79]. When compared to GnRH analogs, progestins show less efficacy in reducing fibroid volume [80]. In a systematic review focused on progestins-based therapy in uterine fibromatosis, an IUD was found to be effective in reducing abnormal uterine bleeding, while oral progestins were not associated either with volume reduction or with improvement of fibroid-related symptoms [81]. 

### 3.6. Androgens

Among androgens, Danazol has previously sparked interest for its potential application in treating symptomatic uterine fibroids. However, the molecular mechanism underlying the reduction in fibroid dimensions remains incompletely understood. One hypothesis suggests that the hypoestrogenic environment leads to the inhibition of DNA synthesis and the induction of apoptosis acting on cell proliferations [82]. While, in the past, some studies highlighted the efficacy of Danazol in reducing fibroid volume (up to 20–25%), and Danazol was thought to be a potential therapeutic option, a subsequent systematic review did not confirm the previous hopes of its efficacy. In fact, the lack of RCTs in this matter weighs against the routine use of Danazol [83] also considering the non-negligible side effects related to its use, which primarily include weight gain, muscle cramps, edema, hot flushes, hirsutism, headaches, depression, skin rash, acne, and androgenic effects, such as deepening of the voice [84].

### 3.7. Aromatase Inhibitors

Aromatase inhibitors, among which Letrozole represents the most studied in relation to the therapy of symptomatic fibromatosis, work by inhibiting the conversion of androgens in estrogens, with an antiproliferative effect on fibroids [85]. The literature presents only RCT when comparing aromatase inhibitors and GnRH analogs, in which the fibroids volume reduction was 46% from the baseline after 12 weeks in the aromatase inhibitors group vs. 32% in the GnRH analog group. A systematic review on their use in fibroid-related symptoms found the effect on volume reduction non-statistically significant, and there is, to this day, no actual evidence to recommend their routine use [86]. 

### 3.8. Selective Estrogen Receptor Modulators (SERMs)

Selective Estrogen Receptor Modulators are non-steroidal molecules that bind the estrogen receptor with agonistic, antagonistic, or mixed agonistic/antagonistic activity, and are characterized by tissue-specificity [87]. Among them, Fulvestrant was studied in comparison with a GnRH analog in an RCT, proving to be less effective both in volume reduction and in control of fibroid-relative bleeding [88]. Raloxifene, also, was studied in the treatment of symptomatic fibromatosis, but data are, to this day, still insufficient to recommend its routine use as a monotherapy [56].

## 4. Minimally Invasive Approaches

The therapeutic management of symptomatic uterine fibromatosis has traditionally been surgical, primarily involving myomectomy or hysterectomy. However, recent years have seen an increasing emphasis on providing more conservative treatment options where possible. Intensive research efforts have focused on developing new conservative therapies aimed at not only preserving the uterus and fertility but also reducing morbidity and shortening recovery times compared to conventional surgical methods. As a result, the development of less invasive alternatives to surgery, such as uterine artery embolization (UAE), radiofrequency ablation (RFA), high-intensity-focused ultrasound ablation (HIFU), microwave ablation (MWA) and irreversible electroporation (IRE), has been promoted.

In 1995, Ravina et al. [89,90] introduced uterine artery embolization (UAE) as a promising alternative for managing symptomatic uterine fibroids. This minimally invasive procedure, performed by interventional radiologists, involves selectively blocking the blood vessels supplying fibroids using angiographic techniques, effectively depriving them of nutrients and causing shrinkage [91]. UAE can also be employed preoperatively to reduce fibroid size, thereby minimizing blood loss during surgery [92]. Studies report a significant fibroid volume reduction post-UAE, ranging from 42% to 83%, with a technical success rate of 96.2% [93,94]. Continued shrinkage of uterine volume is observed as the fibroids degenerate over time [95]. Approximately 95% of patients experience symptom relief and an improved quality of life within a year of the procedure, with only 14.4% requiring further treatment within three years, and 4.6% needing additional surgery [96,97]. Despite these encouraging results, concerns about UAE’s impact on fertility persist. Some studies indicate an increased risk of miscarriage, placental issues, and postpartum hemorrhage, likely due to factors such as endometrial ischemia or unintended ovarian embolization [98,99]. Because of these risks, myomectomy remains the preferred option for women who wish to conceive [100].

Among the various techniques available for radiofrequency ablation (RFA), the transvaginal/transcervical approach stands out for its ability to treat fibroids in an outpatient setting, minimizing patient discomfort and significantly reducing recovery time [101]. RFA operates by delivering high-frequency electrical currents (400 KHz) that generate heat, with temperatures exceeding 65 °C, within the fibroid tissue. This thermal effect induces coagulative necrosis, resulting in irreversible cellular damage to the fibrous tissue and associated blood vessels. The necrotic tissue is subsequently reabsorbed by the body, resulting in substantial fibroid volume reduction—typically between 60–80%—over six to twelve months. The primary goals of RFA treatment are to achieve a significant reduction in fibroid size, potentially making them undetectable in some cases, and to alleviate or completely resolve the symptoms associated with fibroids. Christoffel et al. reported pregnancy outcomes after transcervical fibroid ablation (TFA), noting 36 pregnancies among 28 treated women, resulting in 20 deliveries, with five women conceiving more than once after the ablation, and four achieving pregnancy through assisted reproductive technology (ART) [102,103,104]. 

High-intensity focused ultrasound (HIFU), guided by either ultrasound (US) or magnetic resonance imaging (MRI), has emerged as a promising noninvasive option for treating uterine fibroids, adenomyosis, and other gynecological conditions. This technique focuses US energy on specific points within the fibroid, generating heat between 65 °C and 85 °C, which induces coagulative necrosis without requiring surgical incisions. With a success rate of approximately 90% in reducing fibroid size and alleviating symptoms, HIFU has gained clinical acceptance. Unlike traditional surgery, HIFU does not involve cutting or bleeding, and it generally results in fewer complications [102,103].

Microwave ablation (MWA) is another thermal treatment method that induces cell death by raising tissue temperatures through the generation of electromagnetic waves. This process causes water molecules within the target tissue to oscillate, leading to cellular destruction. Compared to other ablation techniques, MWA achieves higher temperatures more rapidly allowing for a more efficient treatment process and potentially reducing procedure time. Additionally, its enhanced tissue penetration means that larger or more deeply situated fibroids can be effectively targeted, with minimal heat sink effect—where thermal energy is lost to surrounding tissues or blood flow—further enhancing MWA’S precision and efficacy [102,105].

However, thermal ablation procedures, which involve using temperatures exceeding 60 °C to destroy fibroids, carry the risk of thermal damage to surrounding tissues, including nerves, blood vessels and skin due to heat diffusion. A novel approach called irreversible electroporation (IRE) is emerging as a promising alternative for treating solid tumors. Unlike thermal ablation, IRE does not generate high temperatures during treatment, thereby minimizing the risk of collateral thermal injury to adjacent tissues. This nonthermal technique is currently being explored in clinical trials as an experimental therapy for gynecological cancers and may soon be applied to the treatment of fibroids and other solid tumors [102]. 

## 5. Surgical Approach

A myomectomy surgical procedure is an available therapeutic option (Figure 3), especially for fertile-aged women with an unfinished desire for pregnancy who are experiencing myoma-related symptoms, such as pelvic heaviness and pain, heavy menstrual bleeding often leading to anemia, abnormal uterine bleeding, urinary or bowel dysfunctions, and subfertility. The choice of the surgical approach is far from standardized. Various factors may lead to different surgical procedures, including patients’ characteristics like age, desire for pregnancy, and preference, as well as disease-specific factors (size, number, location, related symptoms), and the surgeon’s preferred approach. Abdominal, laparoscopic, robot-assisted, and hysteroscopic myomectomy can be employed, providing a wide range of results in terms of patients’ postoperative symptoms relief, fertility rates, myoma recurrence, and quality of life. Tailoring the diagnostic pathway of each patient is crucial in order to choose the best surgical approach and, consequently, to obtain the best outcomes (Table 2). Last but not least, the risk of symptoms recurrence should be included in the patient’s preoperative evaluation.

### 5.1. Patient Characteristics

#### 5.1.1. Fertility

Desire of pregnancy should be the first factor to consider when dealing with myoma removal. Myoma may be the sole cause of infertility in just 2–3% of the cases, being diagnosed in up to 10% of infertile patients [106,107]. Preservation of the uterus along with myomectomy can be proposed for patients still wishing to conceive, irrespective of their biological age [106]. However, consensus on the positive effect of myomectomy on pregnancy has not yet been reached, especially if patients are stratified according to the myoma’s location [34,108,109,110]. Despite the low-quality evidence in the literature, several hypotheses have been proposed to explain the detrimental effect of myoma on fertility, including distortion of the uterine cavity, hampered sperm transport, deviation and obstruction of the tubal ostia, and chronic endometrial inflammation with altered endometrial receptivity [111]. Patients should be counseled regarding the lack of strong data on the effect of myomectomy on infertility and thoroughly informed that surgical removal might not be the solution and that fibroids are probably not the only cause [112]. Notably, healthcare providers should also discuss the risks with patients affected by uterine fibroids who wish to conceive and have no previous diagnosis of infertility. Myomas in pregnancy are associated with several adverse obstetric outcomes, such as miscarriage, fetal malpresentation, placenta previa, preterm birth, placental abruption, postpartum hemorrhage, and an increased risk of cesarean section in the presence of submucous and intramural fibroids [111,113]. Pre-pregnancy myomectomy is not mandatory but may be advisable in some cases and after careful consideration of all the pros and cons, including the time to conceive after myomectomy that might be estimated at 6 months.

#### 5.1.2. Symptoms

The two most commonly encountered myoma-related symptoms, pain and heavy menstrual bleeding, seem to benefit from the myomectomy surgical procedure [109]. In their retrospective study, Don et al. [114] detected a highly significant reduction in pain and heavy menstrual bleeding VAS scores, by as much as 79% and 89%, respectively. However, other bulk symptoms, such as abdominal pressure and sexual complaints, did not show statistically significant improvements. Similar results are reported by Majak et al. [115], with nearly 80% of patients who experienced pain or vaginal bleeding being satisfied or very satisfied with the myomectomy procedure two years after surgery. Rodriguez-Triana et al. assessed the overall effect of myomectomy on the quality of life of treated patients using the validated Uterine Fibroid Symptom and Health-Related Quality-of-Life Questionnaire (UFS-QOL) [116]. They showed improvement in all areas of quality of life, including energy and mood, concern, activities, control, self-consciousness, and sexual function, even after 12 months and irrespective of the surgical approach employed (laparoscopic vs. open). Candidates for myomectomy should be informed about the beneficial effects of the procedure, which extend beyond the treatment of pain and heavy menstrual bleeding, including improvement of the quality of life and treatment of bulk symptoms.

### 5.2. Myoma Features

In choosing the right surgical approach for myomectomy, knowledge of the myoma characteristics is crucial. The size, number, and location of the fibroids may lead the surgeon to select one technique over another, choosing the most efficient method for the case with apparently less risk of complications. In the systematic review by Cianci et al., both the numbers and the sizes of the fibroids were lower in the group of patients undergoing minimally invasive myomectomy procedures (both laparoscopic and hysteroscopic) compared to the fibroids of patients undergoing abdominal myomectomy [117]. Fibroids removed through the MIS techniques averaged 1 cm less in diameter and were 1.6 less in number than fibroids removed through the open technique. According to this data, there seems to be a trend for the choice of the open approach when greater and/or numerous fibroids must be removed. Several reasons may explain this choice, including the need for high laparoscopic dexterity in complex cases, the experience of the surgeon, and the need of a safe extraction of the myoma without intra-abdominal morcellation. Recently, Casarin et al. have highlighted the importance of size and location of the fibroids through a prospective study investigating myomectomy perioperative complications [118]. According to the univariate analysis, a larger myoma diameter, the intramural location of the myoma, and the opening of the uterine cavity were among the factors associated with postoperative complications. Moreover, greater myoma weight and myoma extending into the broad ligament were significantly associated with blood loss exceeding 1000 mL. Mallick et al. reported greater blood losses and operative times in patients with a large myoma (equal or more than 9 cm of diameter), as well as in those with multiple fibroids (more than four) [119]. More recently, the research group of Ciavattini pointed out the correlation between a higher number of US-diagnosed fibroids (more than four) and an increased intraoperative blood loss (more than 47 mL) [120]. Not surprisingly, the exact morphological diagnosis of a myoma is of utmost importance when planning a myomectomy procedure. Large intramural myomas or those encroaching on the endometrium [121], and multiple fibroids, should cause the surgeon to cautiously ponder the best surgical strategy.

### 5.3. Oncological Risk

Uterine sarcomas are rare and aggressive female genital tract malignancies, accounting for 1% of genital tract tumors. Among the different histotypes, leiomyosarcoma is the most frequently diagnosed, with an incidence of 41–60% [122]. Morcellation of a uterine malignancy inside the abdominal cavity without safety procedures during laparoscopic myomectomies may cause the dissemination of malignant cells [123], potentially leading to the patient’s death. The strength of this association has been recently questioned as morcellation seems not to be the only determinant of malignant cell spillage [124]. Moreover, when endoscopic myomectomy and hysterectomy procedures are accounted together, the overall risk of sarcoma is quite low, standing at 1.5% [125]. However, the US Food and Drug Administration has recently warned surgeons regarding the morcellation of leiomyoma, suggesting that the laparoscopic power morcellation of myomas should be performed only within a tissue-containing system and only in appropriately selected patients [126]. Therefore, the diagnostic workup of a myomectomy should always include the clinical and morphological evaluation of the presumed myoma, supporting the *plausible* exclusion of a uterine sarcoma. An accurate diagnostic pathway is mandatory to guide physicians to a tailored surgical treatment and help them choose the proper surgical route, sometimes even leading to a more invasive procedure such as hysterectomy [122]. Suspicion of malignancy may arise from presenting symptoms, such as abnormal uterine bleeding or tumor growth in postmenopausal women who are not on hormonal replacement therapy [127]. Among biochemical investigations, lactated dehydrogenases may be a useful serum marker, especially when combined with an MRI examination [128]. Moreover, the serum LDH5/LDH1 ratio alone may be indicative of malignancy, but further research is needed to support this tool [129]. According to Raffone et al., pooled sensitivity of ultrasonographic examination and magnetic resonance imaging are 0.76 and 0.90, respectively, while pooled specificity is as high as 0.89 and 0.96 [121]. While ultrasound is deemed a first-line imaging exam in diagnosing suspected uterine sarcoma, MRI assessment is essential as a second-line procedure, especially when the suspicion has not been previously ruled out [130]. Among MRI features suggesting an oncological diagnosis, irregular borders, heterogeneity, a high signal on T2WI intensity, and hemorrhagic and necrotic changes with central non-enhancement, hyperintensity on DWI, and low values for ADC can all be included in the list [128,131]. Considering the high health risks associated with unexpected uterine sarcoma, a pragmatic decision may be required, even leading to hysterectomy in selected cases [132].

### 5.4. Surgeon’s Preference

While the choice of surgical approach is largely driven by myoma features and patient characteristics, it is not yet standardized due to the preferences of both surgeons and patients regarding the surgical technique. Resectoscope myomectomy is reserved for type G0, G1 and G2 myomas, while hysteroscopic G3 myomas treatment is still debated, with promising but scarce data available in the literature [122]. According to the meta-analysis by Giannini et al. comparing the surgical outcomes undergoing laparoscopic or laparotomic myomectomy, laparoscopic myomectomy has a more favorable surgical profile, offering advantages in terms of shorter hospital stays (overall mean difference: 3.12, 95% CI 1.97–4.28, *p* > 0.001), and reduced blood loss (overall mean difference: 0.72, 95% CI 0.22–1.22, *p* < 0.005) [133]. Intraoperative and postoperative complications had no statistically significant differences among the two groups (odds ratio (OR) for intraoperative: 0.89, 95% CI 0.36 to 2.17, *p* = 0.80; OR for postoperative: 1.35, 95% CI 0.72 to 2.52, *p* = 0.35). Interestingly, the analysis did not reveal significant differences in terms of postoperative pain at 24 h from surgery (postoperative analgesia (OR: 0.09, 95% CI −0.23 to 0.41, *p* = 0.58) and obstetric outcomes following myomectomy (OR: 0.78, 95% CI 0.42 to 1.45, *p* = 0.43). Overall, according to these effective data, laparoscopic myomectomy should ideally be preferred when available. However, the application of laparoscopy for myomectomy is debated still nowadays, regardless of the available scientific evidence. Laparoscopic myomectomy is deemed a potentially dangerous surgery [134] especially in unexperienced hands. Kosmacki et al. recently published a retrospective analysis of the learning curve in recently-graduated minimally invasive gynecologic surgeons and showed that young surgeons may initially experience a steep learning curve, with a steady state achieved after 11–23 cases, yet as they take on more difficult cases they may experience a second learning curve that can extend out to 46 cases [135]. Notably, laparoscopic myomectomy is associated with major complications, such as the increased risk of blood transfusion [136], postoperative readmission [137], and undesired hysterectomy [138]. Moreover, performing laparoscopic myomectomy requires not only technical skills but also technological availability, as this procedure is associated with higher costs (OR = 17.61, 95% CI 7.34–27.88) [139]. Lastly, medical comorbidities may preclude the use of laparoscopy, especially when a long duration of operation is contraindicated (OR = 16.10, 95% CI 6.52–25.67) [139]. In these cases, mini-laparotomy may be the preferred technique [140,141]. In their retrospective study comparing 141 mini-laparotomies and 352 laparoscopies, Kumar et al. showed that the open technique may be a safe alternative in selected cases, offering benefits such as shorter operative time (49.3 versus 91.5 min; *p* = 0.003), and reduced estimated blood loss (20 versus 32 mL; *p* = 0.001), probably due to an easier and faster closure of the uterine wound. While the hysteroscopic, laparoscopic, and open technique are still at the top of the list among surgical techniques for myomectomy, robot-assisted laparoscopy has emerged as a viable option in recent years [142]. It offers advantages over the other procedures such as clear three-dimensional vision, manual dexterity with greater instruments flexibility, and a shorter learning curve. A recent meta-analysis by Sheng et al. compared robot-assisted laparoscopic and traditional laparoscopic myomectomy by including 45.702 patients from 15 retrospective studies [143]. Robot-assisted laparoscopic myomectomy showed less intraoperative bleeding (MD = −32.03, 95%CI − 57.24 to − 6.83, *p* = 0.01), a lower risk of blood transfusion (OR = 0.86, 95%CI 0.77 to 0.97, *p* = 0.01), a shorter postoperative hospital stay (MD = −0.11, 95%CI −0.21 to −0.01, *p* = 0.03), fewer conversions to the open technique (OR = 0.82, 95%CI 0.73 to 0.92, *p* = 0.0006), and a lower incidence of postoperative complications (OR = 0.58, 95%CI 0.40 to 0.86, *p* = 0.006), while it was associated with a longer operative time (MD = 38.61, 95%CI 19.36 to 57.86, *p* < 0.0001). Moreover, robot-assisted laparoscopy is intrinsically associated with higher costs due to the concurrent limited spread of the technology, and worse aesthetic results due to the high abdomen skin incisions required for pelvic surgery. Therefore, according to the available data, we can confidently state that there is no standardization, nor should there be, due to the various patient, myoma, and surgeon preferences and indications that drive the choice of technique. The decision to use a specific procedure should be tailored after a thorough evaluation of the patient, the surgeon’s skills, and the surgical setting, and then discussed and agreed upon with the patient.

## 6. Conclusions

The management of a patient with uterine fibromatosis always requires a complex and multidimensional approach, beginning with a thorough assessment of the individual clinical presentation and its potential evolution over time. Advances in diagnostic imaging, particularly ultrasonography and second-level magnetic resonance imaging (MRI), have significantly enhanced our ability to characterize the disease with greater precision. These tools are crucial in guiding treatment decisions and assessing the risk of malignancy and infertility.

Patient counseling is essential in the management of uterine fibroids. Although the presence of fibroids does not inherently cause infertility, and their removal does not ensure a universal improvement in fertility, it is crucial to consider the patient’s desire for pregnancy. A comprehensive discussion of the potential benefits and risks associated with myomectomy is necessary to make informed decisions tailored to the patient’s reproductive goals and overall health.

Pharmacological treatments offer a spectrum of options, ranging from traditional symptomatic hormonal treatments to the latest GnRH antagonists, each providing varying degrees of symptomatic relief. The choice of surgical treatment, which is often necessary to successfully manage the symptoms, must be tailored to the fibroids’ location, number, size, and morphology, as well as the surgeon’s experience and preferences. This can involve minimally invasive approaches like hysteroscopic or laparoscopic myomectomy or more radical procedures such as hysterectomy. Despite the lack of a definitive cure for fibromatosis, the array of available therapeutic strategies allows for considerable mitigation of the condition’s clinical impact. The key to optimizing patient outcomes lies in the personalization of treatment plans, ensuring they align with the patient’s reproductive desires and overall health goals.

The analysis presented underscores the importance of a comprehensive, individualized approach to managing uterine fibromatosis. By integrating detailed diagnostic evaluations with tailored therapeutic strategies, clinicians can better address the complexities of this condition and improve patient care outcomes.

## Figures and Tables

**Figure 1 diagnostics-14-02046-f001:**
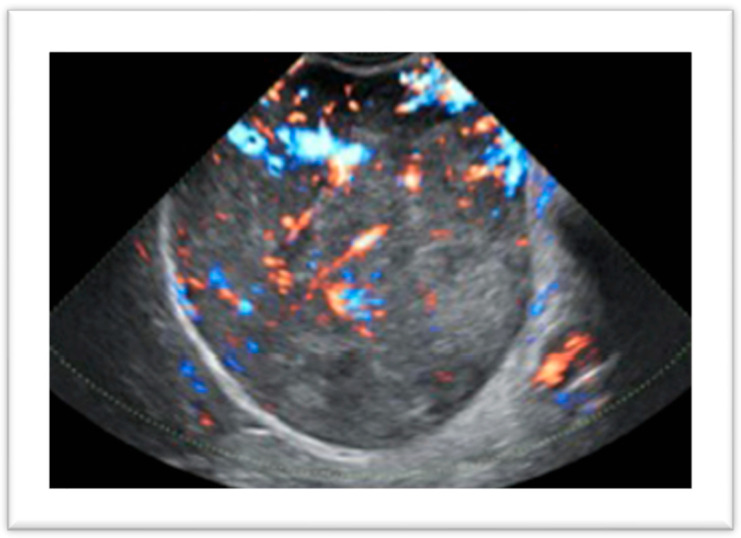
Uterine leiomyoma with atypical vascular features.

**Figure 2 diagnostics-14-02046-f002:**
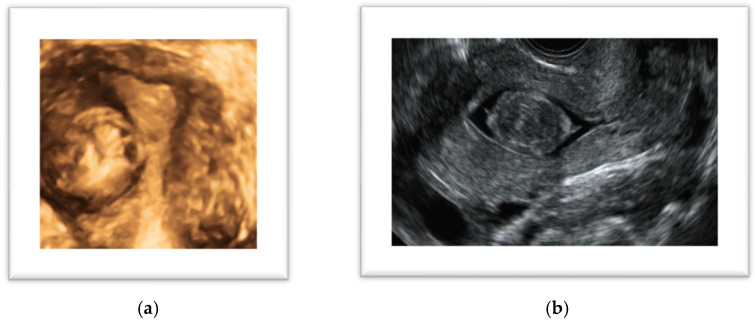
(**a**) Three-dimensional ultrasound coronal view of a uterine fibroid; (**b**) sonohysterography: sagittal view of a G0 (FIGO classification) leiomyoma.

**Figure 3 diagnostics-14-02046-f003:**
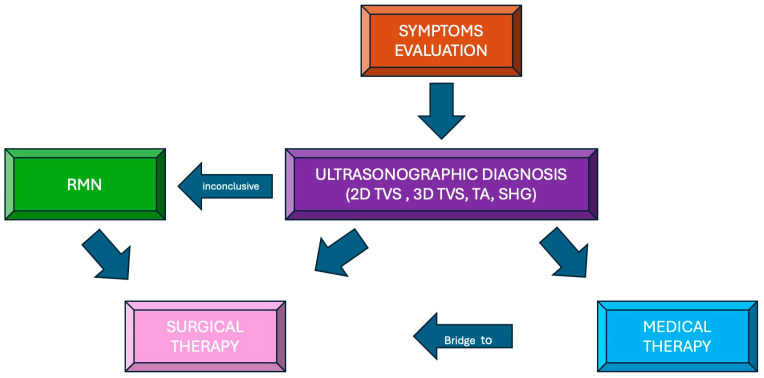
A pragmatic scheme to follow in the evaluation of a patient with uterine leiomyomas.

**Table 1 diagnostics-14-02046-t001:** Summary of the diagnostic tool for the detection of myomas and their main characteristics.

Diagnostic Method	Main Characteristics
Ultrasound	First line imaging technique Appearance of fibroids: Rounded solid lesions Well-defined margins Possible calcifications Internal fan-shaped shadows Peripheral vascularization 3D scans allow accurate location and measurement the myoma by obtaining a coronal view of the uterus
Sonohysterography	Second line imaging technique Appearance of fibroids: As on TVS The distension of the uterine cavity allows precise assessment of the position of the fibroid and identification of the intramural versus intracavitary portion Allows assessment of the size, depth, and vascularization of the entire myoma Possibility of integration with 3D scan
Hysteroscopy	Gold standard for uterine cavity assessment Appearance of fibroids: Solid Smooth protrusion of the uterine wall Direct visualization of the intracavitary portion of the lesion Intramural portion and myometrium free distance to serosa are not assessable Possible “see and treat"
Magnetic Resonance Imaging	Only in complex or doubtful cases Appearance of fibroids: T1 sequences: high T1 signal if non-degenerated fibroids or presence of calcifications T2 sequences: low signal intensity and hypervascularization High T2 signal in presence of cystic degeneration or necrosis Low T2 signal in presence of hyaline degeneration

**Table 2 diagnostics-14-02046-t002:** The main characteristics of medical therapy options (SPRMs: Selective Progesterone Receptor Modulators; AUBs: Abnormal Uterine Bleedings; COC: Combined Oral contraceptives; HMB: Heavy Menstrual Bleeding; LNG-IUS: Levonorgestrel-releasing Intrauterine System; AIs: Aromatase Inhibitors; SERMs: Selective Estrogen Receptor Modulators).

Medical Therapy	Main Characteristics
SPRMs	• antiproliferative effect, proapoptotic effect, reduction in fibrotic activity and reduction in growth factor expression • reducing AUBs and fibroid-related dysmenorrhea • volume reduction both in fibroids and in the uterus • clinical improvement regarding symptoms
GnRH analogues	• determine a hypo-estrogenic, a “temporary menopause” with amenhorrea by the suppression of the pituitary-ovarian • initial stimulation of gonadotropin release ("flare effect") • blocking further fibroid growth • optimal duration of the therapy is 3 to 6 months
GnRH antagonists	• immediate suppression of gonadotropin release without the "flare-up" effect • reduces menstrual blood loss • add-back therapy effectively limited the side effects • slight decrease in fibroid volume
COC	• inhibiting effect on endometrial proliferation, with consequent maintenance of a thin endometrium • not reduce fibroid volume or uterine size • HMB recurs after cessation of treatment → short term efficacy
Oral progestins and LNG-IUS	• causing endometrial atrophy while having a variable effect on fibroids growth • LNG-IUS can reduce significantly both blood loss and uterine volume without any variation on fibroids volume
Androgens	• Hypothesis of molecular mechanism → hypoestrogenic environment leads to inhibition of DNA synthesis and induction of apoptosis acting on cell proliferation • Non-negligible side effects related
Aromatase inhibitors	• inhibiting the conversion of androgens in estrogens, with an antiproliferative effect on fibroids • currently, no strong evidence to recommend routine use of AIs for uterine fibroids
SERMs	• non-steroidal molecules, that bind the estrogen receptor → agonistic, antagonistic or mixed activity, with tissue-specificity • insufficient data to recommend routinary use as monotherapy

## Data Availability

Data availability is not applicable to this article as no new data were created or analyzed in this study.

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
