# Peer review of "Tailoring the Diagnostic Pathway for Medical and Surgical Treatment of Uterine Fibroids: A Narrative Review"

_diagnostics, 2024, doi:10.3390/diagnostics14182046_

Round 1

Reviewer 1 Report

Comments and Suggestions for Authors

A very interesting manuscript entitled “Tailoring the diagnostic pathway for medical and surgical treat-2 ment of uterine fibroids: a narrative review”.

The authors of the mamoscript present an interesting dianostic-terpaeutic analysis on symptomatic uterine myomas.

In my opinion, the paper lacks an analysis of minimally invasive methods of treating symptomatic uterine fibroids - I mean uterine artery embolisation and  focused ultrasound - the text of the manuscript should be supplemented by an analysis of these topics.

In my opinion, it is worth improving the section “ conclusions”  - is written too laconically, while it should highlight the merits of the analysis made by the authors during the editing of this manuscript.

Author Response

RESPONSE TO REVIEWER 1 COMMENTS 

Thank you very much for taking the time to review our manuscript. Please find the detailed responses below. The location of the revised text is referred to the new uploaded version. 

Comment #1: A very interesting manuscript entitled “Tailoring the diagnostic pathway for medical and surgical treatment of uterine fibroids: a narrative review”.

The authors of the manuscript present an interesting diagnostic-terapeutic analysis on symptomatic uterine myomas.

Authors response: We thank the Reviewer for the appraisal. 

Comment #2: In my opinion, the paper lacks an analysis of minimally invasive methods of treating symptomatic uterine fibroids - I mean uterine artery embolization and  focused ultrasound - the text of the manuscript should be supplemented by an analysis of these topics.

Authors response: We thank the Reviewer for the suggestion. We have included, as suggested, in the text the analysis of literature data about minimally invasive approaches, i.e. uterine artery embolization (UAE), radiofrequency ablation (RFA), high-intensity-focused ultrasound ablation (HIFU), microwave ablation (MWA) and irreversible electroporation (IRE).

Location: Lines 438-505, page 11-12.

Text: The therapeutic management of symptomatic uterine fibromatosis has traditionally been

surgical, primarily involving myomectomy or hysterectomy. However, recent years have

seen an increasing emphasis on providing more conservative treatment options where

possible. Intensive research efforts have focused on developing new conservative

therapies aimed at not only preserving the uterus and fertility but also reducing morbidity

and shortening recovery times compared to conventional surgical methods. As a result,

the development of less invasive alternatives to surgery, such as uterine artery

embolization (UAE), radiofrequency ablation (RFA), high-intensity-focused ultrasound

ablation (HIFU), microwave ablation (MWA) and irreversible electroporation (IRE), has

been promoted.

In 1995, Ravina et al. (89-90) introduced uterine artery embolization (UAE) as a

promising alternative for managing symptomatic uterine fibroids. This minimally invasive

procedure, performed by interventional radiologists, involves selectively blocking the

blood vessels supplying fibroids using angiographic techniques, effectively depriving

them of nutrients and causing shrinkage (91). UAE can also be employed preoperatively

to reduce fibroid size, thereby minimizing blood loss during surgery (92). Studies report a

significant fibroid volume reduction post-UAE, ranging from 42% to 83%, with a technical

success rate of 96.2% (93,94). Continued shrinkage of uterine volume is observed as

the fibroids degenerate over time (95). Approximately 95% of patients experience symptom relief and improved quality of life within a year of the procedure, with only

14.4% requiring further treatment within three years, and 4.6% needing additional

surgery (96,97). Despite these encouraging results, concerns about UAE’s impact on

fertility persist. Some studies indicate an increased risk of miscarriage, placental issues,

and postpartum hemorrhage, likely due to factors such as endometrial ischemia or

unintended ovarian embolization (98,99). Because of these risks, myomectomy remains

the preferred option for women who wish to conceive (100).

Among the various techniques available for radiofrequency ablation (RFA), the

transvaginal/transcervical approach stands out for its ability to treat fibroids in an

outpatient setting, minimizing patient discomfort and significantly reducing recovery time

(101). RFA operates by delivering high-frequency electrical currents (400 KHz) that

generate heat, with temperatures exceeding 65°C, within the fibroid tissue. This thermal

effect induces coagulative necrosis, resulting in irreversible cellular damage to the

fibrous tissue and associated blood vessels. The necrotic tissue is subsequently

reabsorbed by the body, resulting in substantial fibroid volume reduction—typically

between 60-80%—over six to twelve months. The primary goals of RFA treatment are to

achieve a significant reduction in fibroid size, potentially making them undetectable in

some cases, and to alleviate or completely resolve the symptoms associated with

fibroids. Christoffel et al. reported pregnancy outcomes after transcervical fibroid ablation

(TFA), noting 36 pregnancies among 28 treated women, resulting in 20 deliveries, with

five women conceiving more than once after the ablation, and four achieving pregnancy

through assisted reproductive technology (ART) (102-104).

High-intensity focused ultrasound (HIFU), guided by either ultrasound (US) or magnetic

resonance imaging (MRI), has emerged as a promising noninvasive option for treating

uterine fibroids, adenomyosis, and other gynecological conditions. This technique

focuses US energy on specific points within the fibroid, generating heat between 65°C

and 85°C, which induces coagulative necrosis without requiring surgical incisions. With a

success rate of approximately 90% in reducing fibroid size and alleviating symptoms,

HIFU has gained clinical acceptance. Unlike traditional surgery, HIFU does not involve

cutting or bleeding, and it generally results in fewer complications (102,103).

Microwave ablation (MWA) is another thermal treatment method that induces cell death

by raising tissue temperatures through the generation of electromagnetic waves. This

process causes water molecules within the target tissue to oscillate, leading to cellular

destruction. Compared to other ablation techniques, MWA achieves higher temperatures

more rapidly allowing for a more efficient treatment process and potentially reducing

procedure time. Additionally, its enhanced tissue penetration means that larger or more

deeply situated fibroids can be effectively targeted, with minimal heat sink effect - where

thermal energy is lost to surrounding tissues or blood flow - further enhancing MWA’S

precision and efficacy (102,105).

However, thermal ablation procedures, which involve using temperatures exceeding

60°C to destroy fibroids, carry the risk of thermal damage to surrounding tissues,

including nerves, blood vessels and skin due to heat diffusion. A novel approach called

irreversible electroporation (IRE) is emerging as a promising alternative for treating solid

tumors. Unlike thermal ablation, IRE does not generate high temperatures during

treatment, thereby minimizing the risk of collateral thermal injury to adjacent tissues.

This nonthermal technique is currently being explored in clinical trials as an experimental

therapy for gynecological cancers and may soon be applied to the treatment of fibroids

and other solid tumors (102).

Comment #3: In my opinion, it is worth improving the section “ conclusions”  - is written too laconically, while it should highlight the merits of the analysis made by the authors during the editing of this manuscript.

Authors response: We thank the Reviewer for the suggestion.  We wrote conclusions about the importance of the analyses in assessing diagnostic pathways and treatment options.

Location: Lines 686-713; Page 16-17

Text: The management of a patient with uterine fibromatosis always requires a complex and

multidimensional approach, beginning with a thorough assessment of the individual

clinical presentation and its potential evolution over time. Advances in diagnostic

imaging, particularly ultrasonography and second-level magnetic resonance imaging

(MRI), have significantly enhanced our ability to characterize the disease with greater

precision. These tools are crucial in guiding treatment decisions and assessing the risk

of malignancy and infertility.

Patient counseling is essential in the management of uterine fibroids. Although the

presence of fibroids does not inherently cause infertility, and their removal does not

ensure a universal improvement in fertility, it is crucial to consider the patients desire for

pregnancy. A comprehensive discussion of the potential benefits and risks associated

with myomectomy is necessary to make informed decisions tailored to the patients

reproductive goals and overall health.

Pharmacological treatments offer a spectrum of options, ranging from traditional

symptomatic hormonal treatments to the latest GnRH antagonists, each providing

varying degrees of symptomatic relief. The choice of surgical treatment, which is often

necessary to successfully manage the symptoms, must be tailored to the fibroids;

location, number, size, and morphology, as well as the surgeon’s experience and

preferences. This can involve minimally invasive approaches like hysteroscopic or

laparoscopic myomectomy or more radical procedures such as hysterectomy. Despite

the lack of a definitive cure for fibromatosis, the array of available therapeutic strategies

allows for considerable mitigation of the conditions clinical impact. The key to optimizing

patient outcomes lies in the personalization of treatment plans, ensuring they align with

the patient’s reproductive desires and overall health goals.

The analysis presented underscores the importance of a comprehensive, individualized

approach to managing uterine fibromatosis. By integrating detailed diagnostic

evaluations with tailored therapeutic strategies, clinicians can better address the

complexities of this condition and improve patient care outcomes.

Reviewer 2 Report

Comments and Suggestions for Authors

The narrative review entitled “Tailoring the diagnostic pathway for medical and surgical treatment of uterine fibroids: a narrative review” presents the different diagnostic and therapeutic methods in the management of fibromas.

Abstract

-              Abstract is too short and concise. I suggest the authors to rewrite it.

Introduction

-              Introduction section is too lengthy. The main issues raised are epidemiology, pathogenesis and symptoms. Introduction should be divided in paragraphs.

-              The topics covered in the review are diagnostic pathway and treatment. These two should be introduced in the introduction.

-              I suggest the addition of a last paragraph that would clearly state the research question and the aim and scope of the review.

Diagnosis

-              I suggest that the section referring to infertility and IVF outcomes should be removed. It is a very hot topic that cannot be covered in a few sentences as a comment in a sonographic evaluation section

-              I suggest the authors to include a table with the main characteristics of each method

Medical Treatment

-              I suggest the authors to include a table summarizing the main characteristics of each treatment

Reference List

-              Correct references 80 and 81

Comments on the Quality of English Language

Moderate editing of English language required.

Author Response

RESPONSE TO REVIEWER 2 COMMENTS 

Comment #1: The narrative review entitled “Tailoring the diagnostic pathway for medical and surgical treatment of uterine fibroids: a narrative review” presents the different diagnostic and therapeutic methods in the management of fibromas.

Abstract

-              Abstract is too short and concise. I suggest the authors to rewrite it.

Authors response: We thank the Reviewer for the suggestion. We have rewritten the abstract as suggested.

Location: Lines 10-21; Page 1

Text: Uterine leiomyomas are the most common benign uterine tumors in women and are often asymptomatic, with clinical

manifestation occurring in 20-25% of cases. The diagnostic pathway begins with clinical suspicion and includes

ultrasound examination, diagnostic hysteroscopy, and, when deemed necessary, magnetic resonance imaging. The

decision-making process should consider the impairment of quality of life due to symptoms, reproductive desire,

suspicion of malignancy, and, of course, the woman’s preferences. Despite the absence of a definitive cure, the

management of fibroid-related symptoms can benefit from various medical therapies, ranging from symptomatic

treatments to the latest hormonal drugs aimed at reducing the clinical impact of fibroids on women’s well-being. When

medical therapy is not a definitive solution for a patient, it can be used as a bridge to prepare the patient for surgery.

Surgical approaches continue to play a crucial role in the treatment of fibroids, as the gynecologist has the opportunity

to choose from various surgical options and tailor the intervention to the patient’s needs. This review aims to

summarize the clinical pathway necessary for the diagnostic assessment of a patient with uterine fibromatosis,

presenting all available treatment options to address the needs of different types of women.

Comment #2:

Introduction

-              Introduction section is too lengthy. The main issues raised are epidemiology, pathogenesis and symptoms. Introduction should be divided in paragraphs.

-              The topics covered in the review are diagnostic pathway and treatment. These two should be introduced in the introduction.

Authors response: We thank the Reviewer for the suggestion. We have divided the introduction into sections as suggested, focusing on the diagnostic pathway and treatment, and as highlighted, we have reduced the sections on epidemiology, pathogenesis and symptoms.

Location: Lines 26-62; Page 1-2

Text: Uterine leiomyomas, or fibroids, are the most common benign tumors affecting women

of reproductive age. While they are clinically apparent in 20–25% of women, histological

diagnoses after hysterectomy suggest a prevalence of up to 70%, with rates as high as

80% among black women by age 50 (1-3). Despite their prevalence, many fibroids

remain undiagnosed due to their asymptomatic nature, potentially leading to an

underestimation of their true impact (4).The substantial prevalence of uterine fibroids

significantly impacts global healthcare expenditures. Research estimates indicate that

annual direct and indirect costs associated with uterine fibroids reach $34.4 billion in the

United States (5).

The pathogenesis of fibroids is multifactorial, involving genetic mutations (e.g., MED12),

hormonal imbalances, and environmental factors that contribute to abnormal cell

proliferation and excessive extracellular matrix deposition (6-10). Fibroids often

overexpress hormonal receptors for estrogen and progesterone, driving their growth in

hormonally favorable environments (11-13). Risk factors for fibroid development include

early menarche, obesity, and chronic stress, while protective factors include multiparity

and the use of combined oral contraceptives (14-20).

Symptomatically, fibroids range from being asymptomatic to causing significant

morbidity, including abnormal uterine bleeding (AUB), pelvic pain, and reproductive

challenges such as infertility and recurrent miscarriage (21-24). These symptoms depend largely on the fibroid’s location—submucosal fibroids, in particular, are more

likely to cause AUB and reproductive issues (25-27).

Given the widespread occurrence of fibroids and their potential impact on quality of life,

accurate diagnosis and tailored management are essential. The diagnostic pathway

often involves clinical examination, imaging techniques like ultrasound and MRI, and

sometimes histological analysis to confirm the diagnosis and plan treatment (9, 26).

Management strategies for fibroids are diverse, ranging from medical therapies that

target hormonal pathways to surgical interventions like myomectomy and hysterectomy,

depending on the size, location, and symptoms of the fibroids (7, 23). This review aims

to provide a comprehensive overview of the clinical pathway for the diagnostic

assessment of patients with uterine fibromatosis. It encompasses a step-by-step

approach starting from initial clinical suspicion to advanced imaging techniques and

diagnostic procedures. Furthermore, this paper presents a detailed evaluation of all

available treatment options, including both conservative and surgical approaches, to

address the diverse clinical presentations and needs of different patient profiles.

Emphasis is placed on individualizing treatment strategies to align with the patient’s

reproductive goals, symptom burden, risk of malignancy, and personal preferences,

thereby optimizing both clinical outcomes and quality of life.

Comment #3:

-              I suggest the addition of a last paragraph that would clearly state the research question and the aim and scope of the review.

Authors response: We thank the Reviewer for the suggestion. We have added a paragraph about clearly stating the research question and the aim and scope of the review, as suggested

Location: Lines 54-62, page 2

Text: This review aims to provide a comprehensive overview of the clinical pathway for the diagnostic

assessment of patients with uterine fibromatosis. It encompasses a step-by-step

approach starting from initial clinical suspicion to advanced imaging techniques and

diagnostic procedures. Furthermore, this paper presents a detailed evaluation of all

available treatment options, including both conservative and surgical approaches, to

address the diverse clinical presentations and needs of different patient profiles.

Emphasis is placed on individualizing treatment strategies to align with the patient’s

reproductive goals, symptom burden, risk of malignancy, and personal preferences,

thereby optimizing both clinical outcomes and quality of life.

Comment #4:

Diagnosis

-              I suggest that the section referring to infertility and IVF outcomes should be removed. It is a very hot topic that cannot be covered in a few sentences as a comment in a sonographic evaluation section

Authors response: We thank the Reviewer for the suggestion. We removed the section as suggested.

Comment #5:

-              I suggest the authors to include a table with the main characteristics of each method

Authors response: We thank the Reviewer for the suggestion. We included the suggested tables.

Comment #6:

Medical Treatment

-              I suggest the authors to include a table summarizing the main characteristics of each treatment

 Authors response: We thank the Reviewer for the suggestion. We included the suggested tables.

Comment #7:

Reference List

-              Correct references 80 and 81

Authors response: We thank the Reviewer for the suggestion. We corrected references 80 and 81.

Round 2

Reviewer 1 Report

Comments and Suggestions for Authors

In its current form the manuscript can be published. I have no comments.

Reviewer 2 Report

Comments and Suggestions for Authors

The authors correctly made the relevant amendments according to reviewers comments. Thus, it can be published in the present form.